# Biological Effects of Green Synthesized Al-ZnO Nanoparticles Using Leaf Extract from *Anisomeles indica* (L.) Kuntze on Living Organisms

**DOI:** 10.3390/nano14171407

**Published:** 2024-08-28

**Authors:** S. K. Johnsy Sugitha, R. Gladis Latha, Raja Venkatesan, Alexandre A. Vetcher, Nemat Ali, Seong-Cheol Kim

**Affiliations:** 1Department of Chemistry, Holy Cross College, Nagercoil, Affiliated to Manonmaniam Sundaranar University, Abishekapatti, Tirunelveli 627012, Tamil Nadu, India; johnsysugitha@gmail.com; 2Department of Chemistry and Research Centre, Holy Cross College, Nagercoil 629002, Tamil Nadu, India; 3School of Chemical Engineering, Yeungnam University, 280 Daehak-ro, Gyeongsan 38541, Republic of Korea; rajavenki101@gmail.com; 4Department of Biomaterials, Saveetha Dental College and Hospitals, SIMATS, Saveetha University, Chennai 600077, Tamil Nadu, India; 5Institute of Biochemical Technology and Nanotechnology, Peoples’ Friendship University of Russia n.a. P. Lumumba (RUDN), 6 Miklukho-Maklaya St., 117198 Moscow, Russia; avetcher@gmail.com; 6Department of Pharmacology and Toxicology, College of Pharmacy, King Saud University, P.O. Box 2457, Riyadh 11451, Saudi Arabia; nali1@ksu.edu.sa

**Keywords:** green approach, antimicrobial, antioxidant, anticancer, anti-diabetic

## Abstract

The synthesis of Al-ZnO nanoparticles (NPs) was achieved using a green synthesis approach, utilizing leaf extract from *Anisomeles indica* (L.) in a straightforward co-precipitation method. The goal of this study was to investigate the production of Al-ZnO nanoparticles through the reduction and capping method utilizing *Anisomeles indica* (L.) leaf extract. The powder X-ray diffraction, UV spectroscopy, Fourier transform infrared spectroscopy, and scanning electron microscopy with EDAX analysis were used to analyze the nanoparticles. X-ray diffraction analysis confirmed the presence of spherical structures with an average grain size of 40 nm in diameter, while UV–visible spectroscopy revealed a prominent absorption peak at 360 nm. FTIR spectra demonstrated the presence of stretching vibrations associated with O-H, N-H, C=C, C-N, and C=O as well as C-Cl groups indicating their involvement in the reduction and stabilization of nanoparticles. SEM image revealed the presence of spongy, spherical, porous agglomerated nanoparticles, confirming the chemical composition of Al-ZnO nanoparticles through the use of the EDAX technique. Al-ZnO nanoparticles showed increased bactericidal activity against both Gram-positive and Gram-negative bacteria. The antioxidant property of the green synthesized Al-ZnO nanoparticles was confirmed by DPPH radical scavenging with an IC_50_ value of 23.52 indicating excellent antioxidant capability. Green synthesized Al-ZnO nanoparticles were shown in in vivo studies on HeLa cell lines to be effective for cancer treatment. Additionally, α-amylase inhibition assay and α-glucosidase inhibition assay demonstrated their potent anti-diabetic activities. Moving forward, the current methodology suggests that the presence of phenolic groups, flavonoids, and amines in Al-ZnO nanoparticles synthesized with *Anisomeles indica* (L.) extract exhibit significant promise for eliciting biological responses, including antioxidant and anti-diabetic effects, in the realms of biomedical and pharmaceutical applications.

## 1. Introduction

The significant rise of nanotechnology has greatly impacted various fields such as science, chemistry, electronics and bio-technology [1]. This is achieved by reducing bulk materials to the nanoscale, resulting in substantial alterations to their structure, physiochemical, optical, and mechanical properties [2]. Various nanomaterials find applications in drug delivery, bio-imaging, gene delivery, nanomedicines, bio-sensing, catalysis, photo-catalytic processes, magnetic resonance imaging, sensor treatment of cancer cells, pharmaceuticals, and memory storage devices [3]. In the previous few decades, there has been a growing need for the synthesis of NPs in various shapes and sizes. Physical methods like microwave processing, solvo-thermal, and ultrasonic processing, along with methods such as hydrothermal, sol-gel synthesis, laser ablation, and lithography, are utilized for the synthesis of metallic nanoparticles [4].

Nanoparticles produced through green synthesis methods are known for their environmentally friendly nature, cost-effectiveness, safety, and biocompatibility enabling the large-scale manufacturing of nanoparticles [5]. Nanoparticles synthesized through chemical or physical methods may pose risks in their respective application fields [6]. The eco-friendly green synthesis method utilizes different plant parts; among these are roots, stems, leaves, flowers, and fruits, which contain phytochemical substances serving as stabilizing and reducing agents during nanoparticles production [7]. The physical and chemical similarities between metallic NPs and bulk metals have attracted interest. These characteristics include low melting point, high surface area, mechanical strength, optical qualities, and magnetic qualities. Metallic NPs have special characteristics like surface plasma resonance and optical characteristics [8]. ZnO nanoparticles stand out among other metal nanoparticles due to their significant applications in the medical field and sensor fabrication. This is explained by their wide band gap, high transmittance, high electron mobility, and significant exciton binding energy [9]. They also have tremendous applications in the medicinal field, such as sunscreen lotion, anti-inflammatory, wound healing, anticancer, antifungal, antioxidant, antibacterial, etc. [10].

Aluminum, a post-transition metal, falls between the boron and carbon families. It is commonly utilized as a dopant element because of its small ionic radius and cost-effectiveness. Therefore, in this study, Al^3+^ is utilized as a dopant element alongside Zinc. By substituting Zn^2+^ ions with the ZnO lattice, electrical conductivity is enhanced by increasing the number of change carriers [11]. ZnO doped with Al exhibits reduced crystallinity and smaller particles size compared to pure ZnO nanoparticles. By exploring the medicinal properties of *Anisomeles indica* (L.) commonly known as catmint and belonging to the Laminaceae plant family [12] which is an erect, camphor-scented, perennial woody shrub, the extract of the plant is taken into account. *Anisomeles indica* (L.) leaf is selected as the novel reducing agent and zinc salt as a precursor for synthesizing aluminum-doped zinc oxide (Al-doped ZnO) nanoparticles due to several key factors related to its unique properties and the advantages it offers in the nanoparticle synthesis process. *Anisomeles indica* (L.) contains various bioactive compounds, such as flavonoids, phenolics, and essential oils. These compounds can act as reducing agents and stabilizers during the synthesis of nanoparticles [13,14]. Their presence can help in controlling the size, shape, and dispersion of the nanoparticles [15]. The plant’s extracts can have strong reducing properties that facilitate the reduction of metal ions (in this case, aluminum and zinc) to form nanoparticles [16].

The plant’s traditional use in medicine might indicate its effectiveness in producing nanoparticles with desirable biological properties, making it a potential candidate for synthesizing nanoparticles with therapeutic or biomedical applications. Numerous important phytochemicals are present in this plant, including alkaloids, terpenoids, arachnoids, and flavones such as iso-ovatodiolide, 4,7-oxycycloanisomelic acid, anisomelic acid, and ovatodiolide [17]. The plant extract is commonly utilized for alleviating conditions like rheumatism, cold, fever, abdominal pain, skin sores, eczema, snake bites, and other ailments when applied topically [18].

This study involved the synthesis of Al-ZnO nanoparticles employing *Anisomeles indica* (L.) leaf extract as a bio-reducing agent. Zinc acetate dihydrate served as the precursor, while aluminum acetate was used as the dopant [19]. The NPs were subsequently classified through XRD UV–Vis, FTIR, and SEM with EDAX FTIR, UV–Vis, SEM, and X-ray diffraction using EDAX techniques. Additionally, their antibacterial, antifungal, antioxidant, anticancer, and anti-diabetic properties were investigated.

## 2. Materials and Methods

### 2.1. Materials

Fresh *Anisomeles indica* (L.) leaves were gathered from Kanyakumari District, Tamil Nadu, India. The chemicals zinc acetate, aluminum acetate, and sodium hydroxide are utilized for analysis and were procured from Merck.

### 2.2. Preparation of Anisomeles Indica Leaf Extracts

In order to remove the dust particles, the plant leaves underwent thorough washing utilizing purified water. The leaves were cleaned, letting them air out in the shade, and then ground using an electric blender. Approximately 10.0 g of the resulting powder was combined with 10 mL of deionized water and heated to 70 °C for 30 min. Whatman filter paper was used to filter the mixture once it had cooled. After that, the filtrate was put into amber bottle and kept cold at 40 °C for later analysis.

### 2.3. Green Synthesis of Al-ZnO Nanoparticles

The green synthesis of Al-ZnO nanoparticles from a solution of *Anisomeles indica* (L.) leaf extract is shown in Figure 1. The preparation process involves adding 2.74 g of zinc acetate dehydrate, 2.53 g of aluminium acetate, and 2.0 g of NaOH to 50 mL of distilled water. Subsequently, 25 mL of plant extract is introduced to the solution with continuous stirring at room temperature for 2 h. The pH is accustomed to 12 by gradually adding 1.0 N NaOH solution. Following 3 h of stirring, a light-yellow solution forms, which is left undisturbed to yield a light-yellow precipitate. We purified the sample with distilled water and ethanol several times. The solution is then dried and calcined at 250 °C for 2 h. *Anisomeles indica* (L.) contain more bioactive compounds such as terpenoids, arachnoids, and flavones such as iso-ovatodiolide, 4,7-oxycycloanisomelic acid, anisomelic acid, and ovatodiolide, alkaloids, triterpenoids, phenolic compound, essential oil, saponins, and tannins. Among these, some of the bioactive compounds degrade when it is heated at 250 °C. But when it is cooled at 4 °C, it retains its property [20]. After purification, the sample is finely powdered and stored in a closed container for further analysis. The color change of the *Anisomeles indica* (L.) leaf extract indicates the synthesis of Al-ZnO nanoparticles when exposed to an Al-Zn (CH_3_COO)_2_·2H_2_O solution. The dried sample undergoes characterization through FTIR, UV–Vis, SEM with EDAX, and XRD method, and biological studies.

### 2.4. Characterization of Al-ZnO Nanoparticles

Using Cu-Kα radiation with wavelength 1.54 Å and angles ranging from 10 to 80°, the XRD pattern is obtained to examine the sample’s crystalline structure [21]. The optical properties of the synthesized nanoparticles were examined through absorption spectroscopy. For observing the optical absorption within the 200–800 nm wavelength range of the Al-ZnO nanoparticles, a UV–visible spectrometer was employed. A spectrometer (Perkin-Elmer Spectrum Two, Shelton, CT, USA) with a 4 cm^−1^ resolution in the 4000–400 cm^−1^ range was used to acquire the sample’s FTIR spectra. A TESCAN VEGA 3SBH scanning microscope (Brno, Czech Republic), was employed to examine the morphology of the Al-ZnO NPs. Elemental details and composition of the nanomaterial were determined through energy dispersive spectrometry. The antimicrobial function of the Al-ZnO NPs was evaluated against three Gram-negative species (*K. pneumoniae*, *E. coli*, and *V. cholerae*) and three Gram-positive (*S. aureus*, *B. substilis*, and *S. metans*) using agar diffusion well variant and agar diffusion disc variant methods [10]. The agar cup technique was employed to examine the antifungal drugs against *Aspergillus flavus* and *Candida albicans*. Using BHT, the 2,2-diphyny; 1-picrylhydrazyl (DPPH) method was utilized to compute the radical scavenging actions [22] serving as the positive control. Cellular metabolic activity and cytotoxicity were calculated using the MTT assay as an indicator of cell viability. The diabetes-prevention action of Al-ZnO NPs was evaluated with α-amylase and α-glucosidase enzymes using a Spectrophotometric Stop method [23].

### 2.5. Microbial Assay

#### 2.5.1. Test Microorganisms

The test organisms in the research were three Gram-positive bacterial organisms such as *S. aureus*, *B. subtilis* (ATCC 25923), and *S. mutans* (ATCC 25922), Gram-negative bacterial organisms such as *K. pneumonea*, *E. coli*, and *V. cholereae*, and the fungi used for the antifungal study, *Candida albicans* and *Aspergillus flavus*.

#### 2.5.2. Zone of Inhibition Assay

Measurements of the inhibitory zone and estimation of the antibacterial characteristics of the produced Al-ZnO NPs against the previously mentioned microorganisms are conducted using agar diffusion well variant and agar diffusion disc variant experiments. The systems were incubated for 24 h at 36 ± 1 °C under aerobic conditions. The bacterial suspension was diluted to 10^8^ CFU/mL (turbidity = McFarland barium sulfate standard 0.5) using sterile physiological solution following a 24 h incubation period in order to prepare for the diffusion and indirect bioautographic tests. The bacterial culture was diluted with BHI broth to a density of approximately 10^9^ UFC/mL for the direct bio-autography test (McFarland standard 3). For an indicator solution to measure the expansion of bacteria for the bacterial growth studies, Sigma-Aldrich offered a 2-(4-iodophenyl)-3-(4-nitrophenyl)-5-phenyl-TTC-(INT) solution in 70% EtOH (2 mg/mL).

#### 2.5.3. Evaluated Methods

Agar diffusion test provided the two types of reservoirs (filter paper disc impregnated with compound test and wells in dishes) and the bioautographic method (Agar diffusion and chromatogram layer) are the techniques used to screen the antibacterial activity of natural products.

#### 2.5.4. Agar Diffusion Well Variant

A sterile cotton swab was used to evenly disperse the bacterial inoculum over a sterile Petri plate MH agar. Concentrations of 100, 80, 60, 40, 20, 10, 5, 2.5, and 1.25 mg/mL were obtained for the experimental materials through nine serial dilutions; four iterations of dilution produced concentrations of 20, 15, 10, and 5 mg/mL for the pure chemicals. Five wells with 7 mm diameter of the agar gel’s pores and spaced 20 mm apart were filled with 50 µL of each of the five experimental samples. The systems were incubated under aerobic conditions for 24 h at 36 ± 1 °C. There was confluent bacterial growth after incubation. The amount of bacteria growth inhibition was expressed in mm. Chloramphenicol (positive control) 30 mg bought in 50 mL were the reference commercial discs utilized.

### 2.6. Antioxidant Studies of Al-ZnO Nanoparticles

#### 2.6.1. DPPH Radical Scavenging Assay

A spectrophotometer was used to measure the absorbance of the incubated sample at 517 nm in order to calculate the percentage of radical scavenging activity (RSA). For the DPPH assay, butylated hydroxytoluene (BHT) was used as a positive control. Briefly, 3.5 mL of 0.1 mm DPPH solution containing various concentrations of sample powder underwent sonication before incubation at 37 ± 2 °C for 30 min under aerobic conditions.

#### 2.6.2. Determination of IC_50_ Value

Using MTT [2,5-diphenyltetrazolium bromide-3-(4,5-dimethylthiazol-2-yl)], the assay was carried out. An amount of 0.02% DMSO (Sigma, St. Louis, MO, USA) was used to dissolve the material. Prior to the experiment, each well of a well plate was planted with five times as many HeLa and L929 cells. After seeding, the cells were exposed to SV-ME and SV-EE for 24 and 48 h, at dosages varying between 0 and 1000 μg/mL. In this assay, cancer cells are exposed to two different treatments (SV-ME and SV-EE) across a range of concentrations and for specified durations. Synthetic variant typically refers to a chemically synthesized compound or drug that is designed to target cancer cells. It might be a new or modified version of an existing drug, created to improve its efficacy or reduce side effects. In the context of anticancer research, a synthetic variant could be a small molecule, peptide, or other types of compounds that are specifically engineered to interact with certain biological targets involved in cancer progression. Metastatic efficacy indicates that this treatment is being evaluated for its ability to address metastatic cancer cells—those that have spread from the original tumor to other parts of the body. The goal is to evaluate the effectiveness and potential toxicity of these treatments. The use of DMSO as a negative control helps ensure that any observed effects are due to the treatments rather than the solvent. This setup allows researchers to determine the potential anticancer activity of SV-ME and SV-EE (Methanolic Extract and Ethanoic Extract) and to understand how these treatments affect cancer cells over time. As the indifferent control, DMSO (0.02%) was used. Following the fifth and sixth treatment days, 20 µL of MTT solution (5 mg/mL MTT diluted in phosphate-buffered saline [PBS]) was added, and it was incubated at 37 °C for four hours. After incubation, 100 µL of 100% DMSO was used to dissolve the purple formazan crystals that had developed. Using a microplate reader, the absorbance of the combination was determined for the sample at 570 nm and the reference at 630 nm. The data are given as means ± SD, with each experiment conducted three times.
(A_t_ − A_b_)/Cell survival = A_t_ × 100 (A_c_ − A_b_)(1)
where A_t_ = Absorbance of test, A_b_ = Absorbance of blank (media), A_c_ = Absorbance of control (cells).

#### 2.6.3. MTT Assay

The MTT [3-(4,5-dimethylthiazol-2-yl)-2,5-dipenyltetrazolium bromide] test (Applichem, Chicago IL, USA) was used to evaluate the cytotoxicity of the experimental material. This assay detects MTT and uses mitochondrial dehydrogenase to transform it into a blue formazan product that signifies mitochondrial activity and cell survival and function. In well plates (Greiner, Frickenhausen, Germany) with 200 µL of growth media, exponentially developing cells were planted in triplicate at a density of 2 × 10^4^ cells/mL. Before adding the extracts, the cells were allowed to incubate for a full day. In order to test plant extracts against six cell lines, a final concentration of 4 mg/mL of the dissolved extracts in 10% DMSO were added to the animal cultures. In an incubator with 5% CO_2_, the cells were cultivated for 72 h at 37 °C. The studies were conducted with final concentrations of 0.1, 0.5, 1.0, 5, and 10 mg/mL and were incubated for 24, 48, and 72 h to assess the responsiveness of the dosage and time. Every well additionally received 10 µL of PBS with 5 mg/mL MTT. Aspiration was used to remove the medium after a 4 h incubation period, and 100 µL of DMSO was used to dissolve the formazan blue crystals that had formed inside the cells. The decrease in MTT was ascertained by using a Thermo Scientific Multiscan Spectrum microplate reader (Thermo Scientific, Waltham, MA, USA) to measure the absorbance at 540 nm. By contrasting the absorbance of treated and untreated cells, the cytotoxicity of the extracts under study was determined. Using the following formula, the relative cytotoxicity to controls was determined:% of Cytotoxicity = [(A_c_ − A_t_)/A_c_] × 100(2)
where A_c_ and A_t_ denote, respectively, the mean absorbance of the control well and the test well.

Propidium iodide is a critical tool in cancer research, not as an anticancer agent but as a diagnostic and analytical reagent. It helps researchers assess cell viability, monitor apoptosis, and analyze cell cycle dynamics, which are essential for evaluating the effects of anticancer treatments and understanding cancer cell biology. Its role is integral to studying and developing new cancer therapies by providing valuable data on how cells respond to various treatments. Examining about 100 cells per group, morphological alterations were evaluated using a 400× magnification fluorescence microscope. Three duplicate tests were conducted to verify the findings.

### 2.7. In Vitro Anti-Diabetic Activity of Al-ZnO Nanoparticles

#### 2.7.1. α-Amylase Inhibition Assay

Using acarbose as the standard, the in vitro anti-diabetic activity of Al-ZnO NPs was assessed by measuring their inhibitory effect on α-amylase. Inhibiting α-amylase activity is crucial for reducing the rate of glucose absorption, as this enzyme plays a key role in carbohydrate hydrolysis. The potential of Al-ZnO nanoparticles as an anti-diabetic agent was examined employing the alpha-amylase assay described by Prasad et al. In this experiment, 50 µL of phosphate buffer containing varying concentrations of the sample solution (10 μL) of α-amylase solution (0.025 mg/mL) was combined with 0.1, 0.2, 0.3, 0.4, and 0.5 mg/mL. Following a 15 min incubation period at 37 °C, 20 μL of P-NPG (5 mm) was included as a substrate into the mixture. Thereafter, 50 μL of 0.1 M Na_2_CO_3_ solution was added and incubated for 20 more minutes. Acarbose was utilized to absorbance of control and a multiplate reader was utilized to quantify the absorbance at 405 nm.
Inhibitory activity (%) = (1 − A_s_/A_c_) × 100(3)
where A_s_ represents the sample’s absorbance and A_c_ represents the control’s absorbance.

#### 2.7.2. α-Glycosidase Inhibition Assay

The potential of Al-ZnO nanoparticles as an antidiabetic agent was evaluated using the α-glycosidase assay. A pre-incubation period of 15 min was conducted at 37 °C with 50 µL of PO_4_ buffer, 10 µL of α-glucosidase, and 20 µL of extract at several concentrations (0.1, 0.2, 0.3, 0.4, and 0.5 mg/mL) for this test. Following that, the mixture was incubated for a further 20 min at 37 °C with the addition of a substrate consisting of 20 μL of P-NPG (P-nitro phenyl-b-D-glucopyranoside) (5 mm). By adding 50 μL of Na_2_CO_3_ solution, the process was stopped. A multi-plate reader was used to measure the absorption of the liberated p-nitrophenol at 405 nm. The experiment was carried out in triplicate, and the percentage blocked of the enzyme was determined using the following formula:Inhibitory activity (%) = (1 − A_s_/A_c_) × 100(4)
where A_c_ is the absorbance of the control and A_s_ is the absorbance when the test medication is present.

## 3. Results and Discussion

### 3.1. Structural Studies of Al-ZnO Nanoparticles Using Anisomeles indica (L.) Leaf Extract

#### 3.1.1. XRD Analysis

The Al-ZnO NPs’ XRD design, which was created using *Anisomeles indica* (L.) leaf extract, is shown in Figure 2. It is evident from the XRD pattern analysis that the hexagonal wurtzite structure of the ZnO-NPs remains constant upon the addition of Al (JCPDS89-0510) [24].

Diffraction peaks were found to broaden as a result of Al doping, which suggested that the size of the nanoparticles had decreased. Strong peaks were seen on this graph, which is consistent with previously published studies, corresponding to 2θ values of angles 31.8°, 34.5°, 47.5°, 56.6°,67.9°, and 69.1° corresponding to (100), (102), (110), (112), and (201), correspondingly, which correspond to different crystal planes [25]. The Debye–Scherrer formula was used to calculate the normal crystallite quantity of the particles [26] and was determined to be 40 nm. The reduction in crystalline size is evident when doping ZnO with aluminium attributed to the lesser ionic radius Al^3+^ contrasted with the Zn^2+^ ions [27].

#### 3.1.2. UV–Vis Studies

The plant’s secondary metabolites aid in the decrease of ‘Zn’ ions in the solution to ZnO [28]. This is supported by the UV–Vis amalgamation spectrum of Al-ZnO NPs synthesized with *Anisomeles indica* (L.) leaf extract, as depicted in Figure 3. Analysis of the absorption spectrum of the ZnO doped with the Al sample reveals the presence of an absorption band below 360 nm, attributed to a slight hypsochromic shift in the absorption peak of the ZnO NPs with Al (approximately 360 nm). Therefore, the Al-ZnO NPs with Al-doped exhibit potential for optical filtering applications [29], attributed to the enhanced band gap absorption below 360 nm resulting from the presence of polyphenol compounds.

The band-gap energy is determined by plotting Tauc’s plot, revealing a direct band gap value of 3.62 eV [30] [refer to Figure 4], attributed to the substitution of Al^3+^ for Zn^2+^. This phenomenon, recognized in the research as the Burstein–Moss result [31], is observed as evidence of the incorporation of Al^3+^ in the ZnO lattice [32]. It can also be utilized as a photosensitive material for UV photon detection due to its broad band gap [33].

#### 3.1.3. FTIR Analysis

The structure and emergence of the functional groups contained in the produced NPs can be determined using FTIR analysis. The Al-ZnO NP’s FTIR spectra prepared with *Anisomeles indica* (L.) leaf extract, as illustrated in Figure 5, indicate that the formation of these nanoparticles is a result of interactions involving phenolic groups, alkenes, alkynes, terpenoids, and flavonoids. The peak observed at 3811 cm^−1^ is attributed to O-H stretching, while the peak at 3441 cm^−1^ corresponds to N-H stretching of amines. The aromatic ring’s C=C stretching is visible at the apex at 1558 cm^−1^. This has a connection to the anhydride acid. The peak at 1573 cm^−1^ indicates that amines are present. The peak at 1496 cm^−1^ represents the C-H stretching of alcohol, while the peak at 1411 cm^−1^ represents the C-N stretching of alkanes. The presence of aromatic amines is indicated by the peak seen at 1360 cm^−1^. The peak at 1081 cm^−1^ is connected with alcohol C-O stretching, while the peak at 786 cm^−1^ is associated with Si-C bond stretching. Moreover, the peak at 643 cm^−1^ illustrates the C=C stretching of alkanes, while the peak at 524 cm^−1^ illustrates the C-Cl stretching of halogens. The combined spectrum of these peaks shows the presence of different biochemical moieties, such as carboxylic compounds found in secondary metabolites like flavonoids with –OH groups and phenolic compounds with –OH functional groups, as well as acids, anhydrides of acids, alcohols, alkanes, alkyl halides, unsaturated hydrocarbons, and amines.

#### 3.1.4. Morphological Studies

SEM was utilized to assess the morphologies on the surface of biosynthesized Al-ZnO NPs with the findings illustrated in Figure 6A–C. The SEM images show the spongy, hexagonal and cubical porous agglomerated nanoparticles with crystalline size ranging from 10 to 50 nm. The particles tend to aggregate due to increased capping capacity, which aids in stabilizing the nanoparticles. The polarity and electrostatic attraction of ZnO NPs cause this agglomeration. The introduction of Al into ZnO diminishes the zinc interstitials for charge compensation, leading to inhibited ZnO grain growth and reduced crystallinity [34]. Figure 6D illustrates the Al-ZnO NP’s EDAX spectra, which show compositions of Al (19.95%), Zinc (27.90%), and Oxygen (52.15%), and confirms the existence of these elements in the proper ratios and free of contaminants.

The TEM images confirm the mixture of spongy, hexagonal, and cubical porous agglomerated nanoparticles as shown in Figure 7A–C. Based on the findings of this investigation, the mean particle size was determined to be 14.54 nm, as illustrated in Figure 7C. The selected area electron diffraction (SAED) of Al-ZnO nanoparticles is shown in Figure 7D. The diffraction rings indicate that the prepared samples are the polycrystalline nature of Al-ZnO. The overall structure suggests the presence of a few small crystal structures within the nanoparticles, which leads to the polycrystalline nature observed; however, specific areas might have clean spots.

### 3.2. Biological Studies of Al-ZnO Nanoparticles

#### 3.2.1. Antibacterial Activity

The sample of *Anisomeles indica* (L.) exhibits varying degrees of inhibitory effects against *K. pneumoniae*, *E. coli*, *V. cholerae*, *S. aureus*, *B. subtilis*, and *S. mutans* at an absorption of 10 mg/mL. Table 1 presents the antibacterial results. In this study, the highest inhibitory effect on antibacterial activity, with a zone of inhibition measuring 4.01 ± 0.01, was found to be anti-*E. coli* at an absorption of 10 mg/mL. Figure 8 shows the antibacterial activity of Al-ZnO nanoparticles using *Anisomeles indica* (L.) leaf extract. The second largest zone of inhibition was observed in the same Gram-negative bacteria against *V. cholerae* (4.04 ± 0.03), with *K. pneumoniae* showing the subsequent value of 2.45 ± 0.01. At a dosage of 5 mg/mL, however, only slight antibacterial effects were noted, with values of 1.65 ± 0.02 mm against *K. pneumoniae* and 2.01 ± 0.45 mm against *V. cholerae*. Additionally, the dominant zones of inhibition were analyzed in the tested Gram-positive organisms against *S. mutans* (5.74 ± 0.01), *S. aureus* (2.74 ± 0.05), and *B. subtilis* (1.62 ± 0.01). Notably, an *Anisomeles indica* (L.) sample demonstrated stronger antimicrobial action directed against the Gram-negative bacteria *E. coli* compared to the other organisms tested. Interestingly, its inhibitory activity against the same organism is expressed akin to the positive control of the antibiotic disc gentamicin.

The present result clearly shows that Al-ZnO NPs employing the leaf extract of *Anisomeles indica* (L.) are potently effective against the clinical pathogens of Gram-positive and Gram-negative organisms. Whenever the absorption of the sample rises, the maximum zone of inhibitory activity is also noticed for the six different clinical pathogens. However, the investigation on the antimicrobial activity of Al-ZnO NPs showed that the antibacterial activity of smaller particles is greater than that of larger molecules. The ionic radius of aluminum is less than that of zinc, which accounts for the size reduction, synergistic effects, and construction of ROS in Al-ZnO samples. A larger surface area-to-volume ratio of smaller Al-ZnO nanoparticles facilitates easier cell membrane penetration. It dissolves quickly and releases harmful metal ions (Al^3+^ and Zn^2+^) when it breaks down between bacterial cell walls. XRD outcomes also denote that the introduction of Al^3+^ ions into the ZnO lattice results in substantial size reduction, thereby enhancing antibacterial activity. Surface morphology research verifies that the Al-ZnO nanoparticles’ abrasive surface damages the bacterial cell membrane, boosting their antibacterial activity. Reactive oxygen species (ROS) produced when nanoparticles interact with pathogens under investigation cause oxidative stress and punctures in the bacterial cell wall membrane, which eventually cause malfunction and rupture. Improving the produced nanoparticles’ inhibitory qualities also depends on this oxidative process. The antibacterial efficaciousness of the Al-ZnO nanoparticles against the pathogens under investigation is markedly boosted by the combined effects of size reduction, surface flaws, abrasive surface roughness, and ROS production.

#### 3.2.2. Antifungal Activity

This study shows that *Aspergillus flavus* (1.89 ± 0.01) followed by (0.78 ± 0.04) in 5 mg/mL has highest antifungal activity with the standard drug amphotericin B viz and no effects are seen in the fungus *Candida albicans*. This result clearly shows that whenever the concentration of the sample increases, the antifungal activity also noticeably increases. The antifungal results are tabulated in Table 2. Al-ZnO nanoparticles synthesized in this study are of 40 nm particle size distribution which has maximum potential and high surface area, indicating the better antimicrobial activity against the tested pathogenic fungal organisms.

#### 3.2.3. Antioxidant Studies

Table 3 presents the antioxidant activity values of green synthesized Al-ZnO nanoparticles. The test samples’ capacity to scavenge radicals against stable 2,2-diphenyl-1-picrylhydrazyl was determined using Brand’s technique. As determined by the DPPH radical scavenging experiment, DPPH reacts with an antioxidant molecule that may donate hydrogen, lowering DPPH through in vitro antioxidant activity. DPPH is responsible for the absorbance at 517 nm with the occurrence of decoloration and subsequently the antioxidant activity of the newly synthesized compound was compared with standard antioxidant BHT. The concentration and percentage of inhibition of the synthesized compounds are given in Table 3. The results of antioxidant potential of the compounds are classified into excellent (IC_50_ = 10–25 µM), good (IC_50_ = 25–40 µM) and moderate (IC_50_ = 40–55 µM) for proving that the synthesized compound possesses excellent antioxidant property (IC_50_ = 23.52) when compared with the standard BHC. When the concentration of the sample increases, the IC_50_ value also increases and hence this sample is said to have excellent antioxidant activity. Furthermore, the NPs connect with phytochemicals through the electrostatic attraction between the negatively charged bioactive molecules in plant extracts, which enhances bioactivity in a synergistic way.

#### 3.2.4. Anticancer Studies

*Anisomeles indica* (L.) was used to assess the Al-ZnO nanoparticles (NPs) in vitro anti-cancer impact, and the results showed a notable amount of activity against HeLa cell lines. The percentage of inhibition ranged from 70.76% to 80.39% at concentrations of 5 µg/mL to 10 µg/mL, indicating a dose-dependent response to the anticancer action (see Figure 9). Significantly, the efficacy of Al-ZnO NPs with *Anisomeles indica* (L.) was similar to that of propidium iodide, a standard anticancer agent. Furthermore, the IC_50_ value, or concentration at which Al-ZnO NPs inhibit 50% of cancer cell growth, was resolved to be 23.61 µg/mL. The obtained IC_50_ value was modest when compared to the positive control, propidium iodide, which had an IC_50_ value of 9.43 µg/mL. Table 4 illustrates the potent anticancer activity of Al-ZnO NPs with *Anisomeles indica* (L.). Microscopic images of HeLa cells treated with various concentrations (5 µg/mL and 10 µg/mL) of Al-ZnO nanoparticles from *Anisomeles indica* (L.) are presented in Figure 9B or Figure 9C). Flow cyclometric analysis of apoptosis induced by Al-ZnO NPs using *Anisomeles indica* (L.) is depicted in Figure 10. Al-ZnO NPs have an anticancer effect on cancer cells, which may be explained by processes such increased intracellular ROS buildup. Highly reactive substances called ROS are known to cause DNA damage, oxidative stress, disruption of cellular functions, and eventually cell death in cancer cells [35]. The NPs may also cause apoptosis, a process of programmed cell death that is essential for preserving cellular homeostasis and removing faulty or damaged cells [36].

In Figure 10, using flow cytometric technique in HeLa cells (a cell line used to treat cervical cancer with Al-ZnO nanoparticles using *Anisomeles indica* (L.) leaf extract at various concentrations is depicted. The results display cells categorized into necrosis (Q1), delayed apoptosis (Q2), viable cells (Q3), late apoptosis previous apoptosis (Q4), contrast to the control.

### 3.3. Anti-Diabetic Studies of Al-ZnO Nanoparticles

#### 3.3.1. α-Amylase Activity

Research is advancing towards the nanomedicine approach, which holds promise as a viable alternative to insulin for managing diabetes. Literature reviews indicate that ZnO is recognized for its anti-diabetic properties [37]. Preventing the activity of α-amylase helps decrease the rate of glucose absorption, as this enzyme plays a crucial role in carbohydrate hydrolysis. The current investigation into the synthesized Al-ZnO nanoparticles as potential α-amylase inhibitors has demonstrated their efficacy as anti-diabetic agents. Table 5 displays the percentage of prevention of α-amylase activity by *Anisomeles indica* (L.). The peak activity was observed in *Anisomeles indica* (L.), exhibiting 75.45% inhibition at the highest concentration and an IC_50_ value of 0.421 mg/mL. The enhanced suppressive effect of *Anisomeles indica* (L.) is attributed to the synergistic action of biomolecules from the leaf extract. The percentage of inhibition decreased in a concentration-dependent manner. It is noted that at lower concentrations, *Anisomeles indica* (L.) exhibits even greater inhibitory potential compared to acarbose, the standard used. The results clearly demonstrate that inhibitory activities increase with higher concentrations of nanoparticles and decrease vice versa [38].

#### 3.3.2. α-Glucosidase Activity

Nanoparticles synthesized from *Anisomeles indica* (L.) demonstrated superior α-glucosidase inhibition activity. The highest concentration exhibited a maximum inhibition of 34.98%, and inhibition percentages ranged from 34.98% to 13.70% across different concentrations. In vitro studies of green synthesized Al-ZnO nanoparticles showed significant α-glucosidase inhibitory activity, resulting in delayed carbohydrate digestion (see Table 5). Depending on the outcomes obtained, it can be concluded that utilizing Al-ZnO nanoparticles synthesized with plant extracts would be highly beneficial in decreasing the rate of carbohydrate digestion and absorption, thus contributing effectively to the management of diabetes.

## 4. Conclusions

The current research encompasses the synthesis of Aluminium with ZnO and *Anisomeles indica* (L.) leaf extract utilizing the co-precipitation method. The resultant nano powder was thoroughly characterized through XRD, UV–Vis, FTIR, SEM with EDAX analyses, confirming their nano structural properties. These synthesized particles were subsequently subjected to a comprehensive array of testing activities, including antibacterial, antifungal, antioxidant, anticancer, and anti-diabetic evaluations. The antibacterial effectiveness of the produced nanoparticles was assessed by the utilization of the agar diffusion disc and well methods. Significant antibacterial activity was strikingly shown against a variety of bacterial species, among these *K. pneumoniae*, *E. coli*, *V. cholerae*, *S. aureus*, *B. substilis*, and *S. mutants*, with the highest inhibitory effect noted against *E. coli* (4.01 ± 0.02). In parallel, the antifungal activity of the nanoparticles was appraised using the agar cup method, demonstrating notable efficacy against fungal species such as *Aspergillus flavus* and *Candida albicans*, with the most pronounced inhibition recorded against *Aspergillus flavus* (1.89 ± 0.01). Additionally, the synthesized NPs’ antioxidant capacity was evaluated using Brand’s technique, which revealed higher activity with an IC_50_ value of 23.52 when compared to normal ascorbic acid. Additionally, the MTT assay was utilized to evaluate the NPs’ anticancer activity, and the outcomes revealed an IC_50_ value of 23.61 µg/mL, indicating that the particles may be able to stop the development of cancer cells. Notably, this value was comparable to the positive control propidium iodide. The assessment of anti-diabetic action through α-amylase and α-glycosidase inhibition assays revealed promising findings. Significant α-amylase inhibition was observed (75.45% at the highest concentration) with an IC_50_ value of 0.421 mg/mL. Additionally, α-glycosidase inhibition activity showed a maximum inhibition of 34.98% at the highest concentration, with an IC_50_ value of 0.308 mg/mL. Conclusively, this research underscores the multifaceted potential of synthesized doped nanoparticles in diverse biological and biomedical applications, thus advocating for their exploration across various domains and laying a robust foundation for future research endeavors in this burgeoning field.

## Figures and Tables

**Figure 1 nanomaterials-14-01407-f001:**
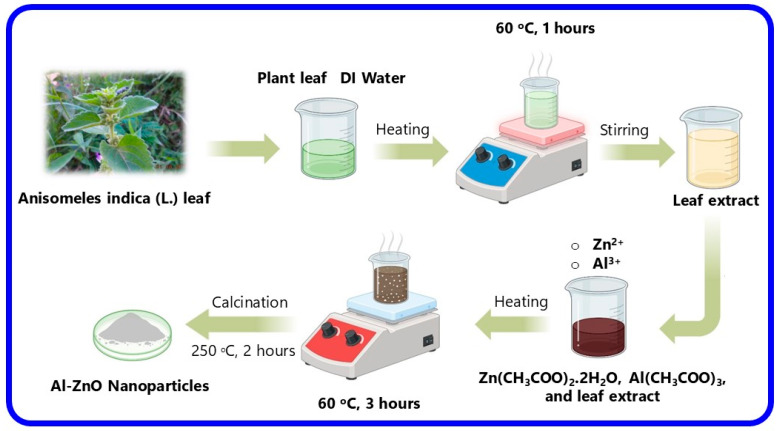
Schematic diagram of Al-ZnO nanoparticles using *Anisomeles indica* (L.) leaf extract.

**Figure 2 nanomaterials-14-01407-f002:**
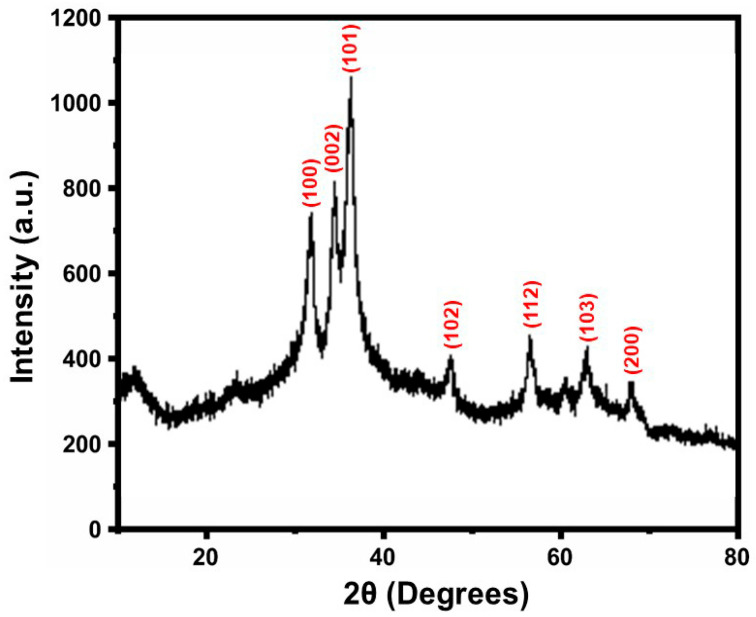
XRD patterns of the Al-ZnO nanoparticles.

**Figure 3 nanomaterials-14-01407-f003:**
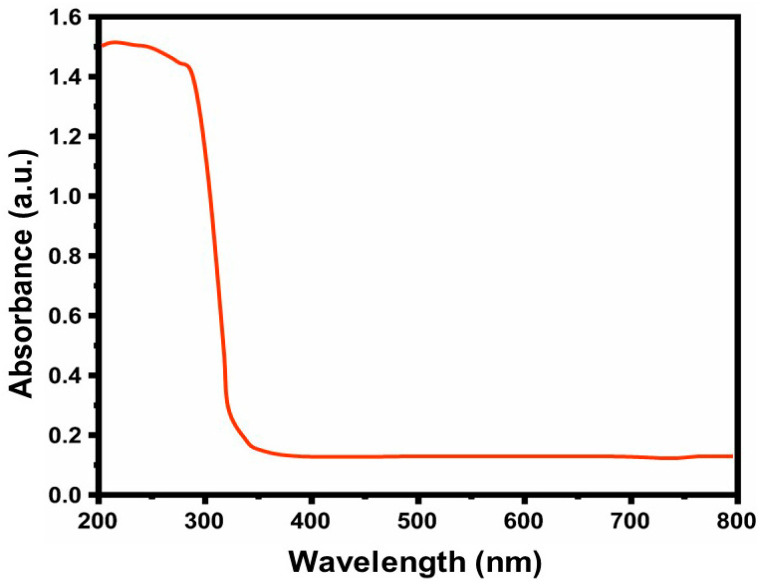
UV–Visible spectrum of the Al-ZnO nanoparticles.

**Figure 4 nanomaterials-14-01407-f004:**
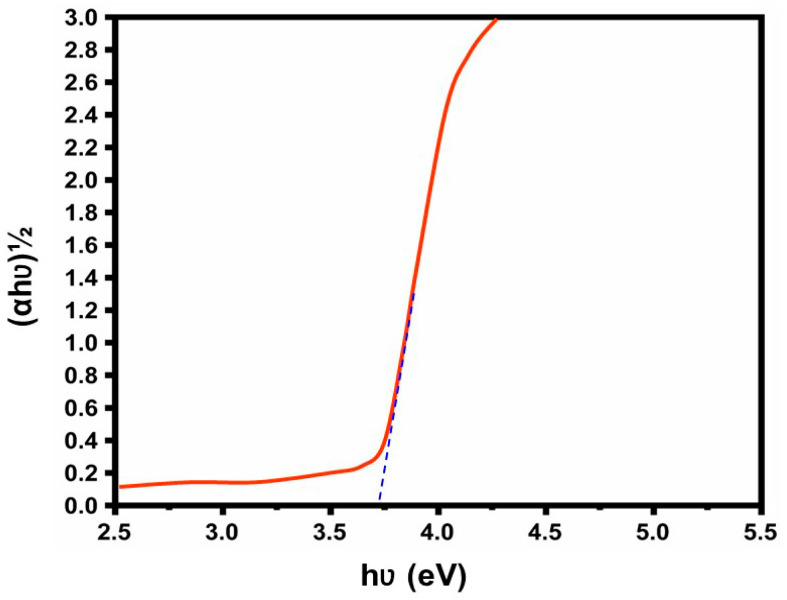
Tauc’s plot of the Al-ZnO nanoparticles.

**Figure 5 nanomaterials-14-01407-f005:**
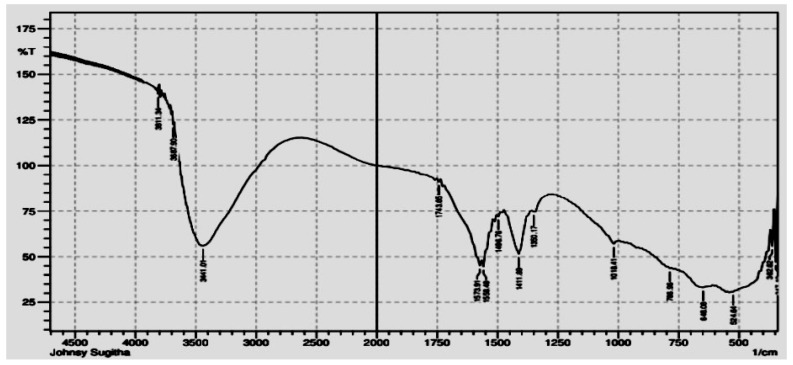
FTIR spectrum of the Al-ZnO nanoparticles.

**Figure 6 nanomaterials-14-01407-f006:**
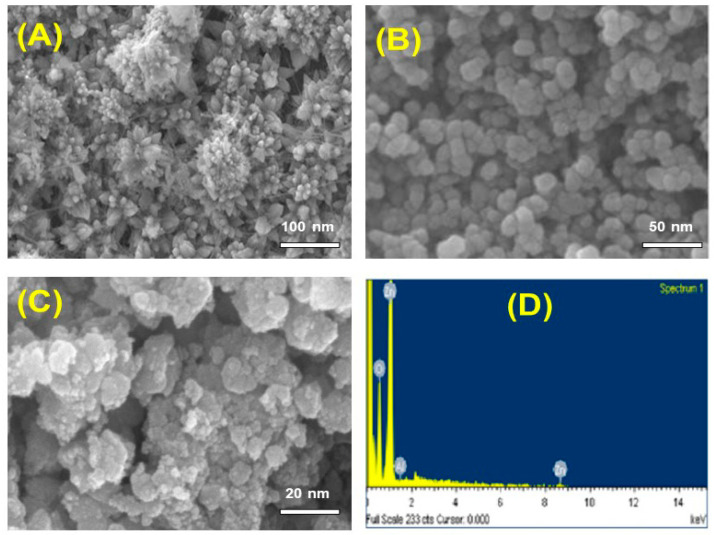
(**A**–**C**) SEM images, and (**D**) EDAX spectrum of Al-ZnO nanoparticles.

**Figure 7 nanomaterials-14-01407-f007:**
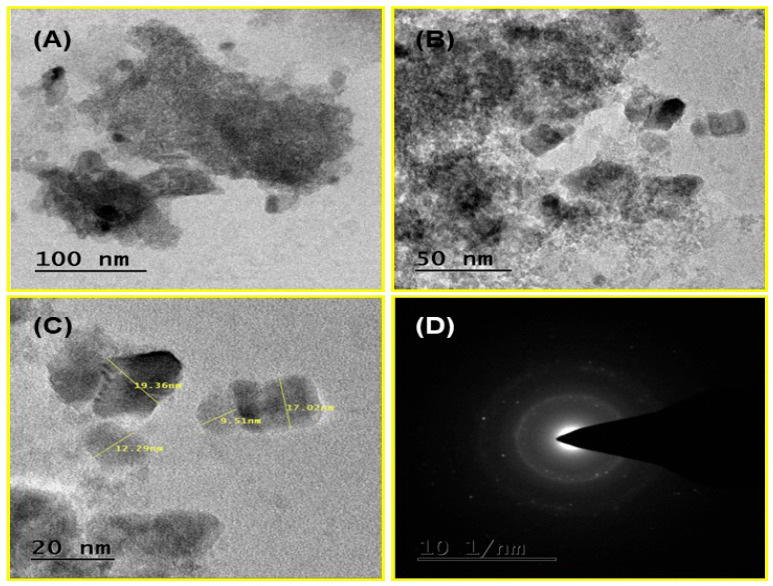
(**A**–**C**) TEM images, and (**D**) SAED image of Al-ZnO nanoparticles.

**Figure 8 nanomaterials-14-01407-f008:**
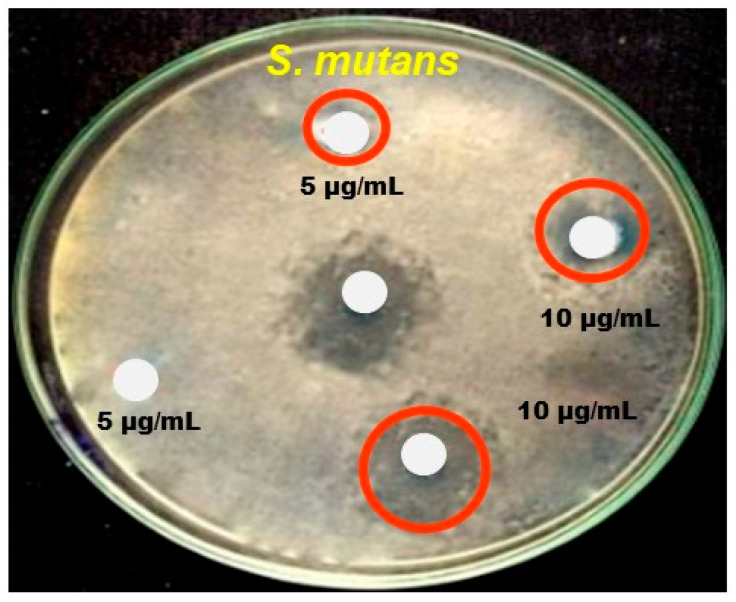
Antibacterial activity of Al-ZnO nanoparticles.

**Figure 9 nanomaterials-14-01407-f009:**
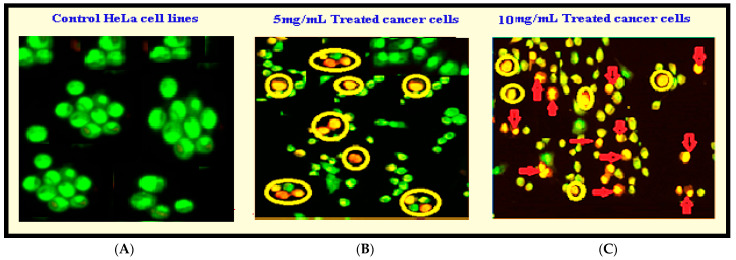
Microscopic images of cells treated for 24 h at various doses; (**A**) HeLa cells treated with different amounts of Al-ZnO NPs (5 µg/mL); (**B**) and 10 µg/ML; (**C**) from *Anisomeles indica* (L.) and the reference standard propidium iodide.

**Figure 10 nanomaterials-14-01407-f010:**
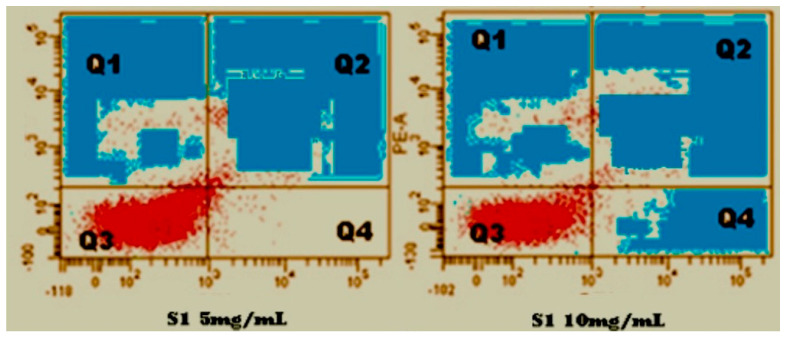
HeLa cells treat cervical cancer with Al-ZnO nanoparticles with various concentrations *using Anisomeles indica* (L.) leaf extract.

**Table 1 nanomaterials-14-01407-t001:** Antibacterial results of the synthesized Al-ZnO nanoparticles (NPs) using the plant *Anisomeles indica* (L.).

Name of the Organisms	Zone of Inhibition (mm-cm)		Positive Control
	5 mg/mL	10 mg/mL	
*K. pneumonea*	1.64 ± 0.21	2.45 ± 0.01	4.63 ± 0.11
*E. coli*	3.17 ± 0.02	4.01 ± 0.02	2.10 ± 0.01
*V. cholera*	2.00 ± 0.45	4.40 ± 0.02	-
*B. subtilis*	1.62 ± 0.21	2.32 ± 0.02	3.25 ± 0.91
*S. aureus*	1.92 ± 0.25	2.74 ± 0.05	-
*S. mutans*	3.00 ± 0.06	5.74 ± 0.07	4.22 ± 0.03

**Table 2 nanomaterials-14-01407-t002:** Antifungal activity of Al-ZnO nanoparticles.

Name of the Organisms	Zone of Inhibition (mm-cm)		Positive Control
	5 mg/mL	10 mg/mL	
*Aspergillus flavus*	0.78 ± 0.02	1.89 ± 0.01	2.00 ± 0.10
*Candida albicans*	-	-	3.22 ± 0.15

**Table 3 nanomaterials-14-01407-t003:** Antioxidant potential of the experimental sample.

S. No	Concentration (mg/mL)	IC_50_ Value (µM)
1.	0.2	18.00 ± 0.10
2.	0.4	20.13 ± 0.37
3.	0.6	24.11 ± 0.71
4.	0.8	30.72 ± 1.52
5.	1.0	41.58 ± 1.23
6.	IC_50_ value mg/mL	23.52 ± 0.03

**Table 4 nanomaterials-14-01407-t004:** Percentage of cell viability for Al-ZnO nanoparticles.

S. No	Concentration (mg/mL)	IC_50_ Value (µM)	Inhibition (%)	IC_50_ mg/mL
1.	1.0	0.411 ± 0.02	32.65	23.61
2.	2.5	0.155 ± 0.02	59.65	
3.	5.0	0.226 ± 0.03	70.76	
4.	10.0	0.213 ± 0.03	80.39	

**Table 5 nanomaterials-14-01407-t005:** α-amylase and α- glucosidase inhibitory effect of green synthesized Al-ZnO nanoparticles (NPs) using *Anisomeles indica* (L.) plant extract.

Concentration (mg/mL)	Absorbant Value of α-Amylase	Absorbant Value of α-Glucosidase	Average	Enzyme Activity (mg/mL)	
				α-Amylase	α-Glucosidase
0.2	18.65	19.07	18.86 ± 0.40	36.38	13.70
0.4	20.13	20.28	20.25 ± 0.20	39.06	18.72
0.6	24.11	23.11	23.61 ± 0.71	45.55	21.07
0.8	30.72	31.08	30.90 ± 0.28	59.61	28.98
1.0	39.58	38.64	39.11 ± 0.37	75.45	34.98

## Data Availability

Data will be made available on request.

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
