# Peer review of "Biological Effects of Green Synthesized Al-ZnO Nanoparticles Using Leaf Extract from Anisomeles indica (L.) Kuntze on Living Organisms"

_nanomaterials, 2024, doi:10.3390/nano14171407_

Round 1

Reviewer 1 Report

Comments and Suggestions for Authors Regarding to Manuscript ID: nanomaterials-3148814
Type of manuscript: Article
Title: Biological Effects of Green Synthesized Al-ZnO Nanoparticles Using Leaf Extract from Anisomeles indica (L.) Kuntze on Living Organisms
Authors: S. K. Johnsy Sugitha, R. Gladis Latha *, Raja Venkatesan, Alexandre
A. Vetcher, Nemat Ali, Seong Cheol Kim *

The authors synthesized some Al-ZnO nanoparticles using a vegetal extract from Anisomeles indica. Then, the resulted particles were physico-chemical and biological characterized.  

Q1: Are you sure was heated to 700 °C?

“Preparation of Anisomeles indica leaf extracts . In order to remove the dust particles, the plant leaves underwent thorough utilizing purified water for washing. The leaves were cleaned, letting it Air out in the shade and then ground using an electric blender. Approximately 10.0 grams of the resulting powder was combined with 10 mL of de-ionized water and heated to 700 °C for 30 minutes. Whatman filter paper was used to filter the mixture once it had cooled. After that, the filtrate was put into amber bottle and kept cold at 40 °C for later analysis.”

Q2: The authors stated the following:

“The solution is then dried and calcined at 250 °C for 2 hours. After purification, the sample is finely powdered and stored in a closed container for further analysis.”

How do they purify the sample?

Q3: In the following is it about Al-NP’s ?

“A variety of hazardous gram-positive (+Ve) and gram-negative bacteria (-Ve) as well 147 as dangerous fungi were tested for the antibacterial and antifungal qualities of green produced Al-NP’s.”

Q4: Please revise it. Seems confuse; incubation of 8 h or 24 h?

“Cultures were incubated for 8 hours, and after 24 hours at 36 ± 1 °C, they were examined for purity.”

Q5: Please revise this phase:

“The bacterial culture was diluted with BHI broth to a density of approximately 109 UFC/mL for the direct bio autography test (McFarland standard 3). indicator solution to measure the expansion of bacteria for the bacterial growth studies, 163 Sigma-Aldrich offered a 2-(4-iodophenyl)-3-(4-nitrophenyl)-5-phenyl-TTC-(INT) solution 164 in 70% EtOH (2 mg/mL).”

Q6: Explanations for SV-ME, SV-EE, P-NPG, BHI and others abbreviations should be introduced when appears for the first time in text.

Q7: The meanings of Ac and At must be reversed:

“Where, Ac and At denote, respectively, the mean absorbance of the test well and the control well.”

Q8: Where is the Fig 3(b)?

“The band-gap energy is determined by plotting Tauc’s plot, revealing a direct band gap value of 3.4eV [26] [refer to Fig. 3(b)], attributed to the substitution of Al3+ for Zn2+ .”

Q9: The resolution of TEM images is quite low, so I am not convinced that the particles are spherical.

Q10: The authors state in the text that Table 1 presents the antibacterial results of the plant (Anisomeles indica), not of the synthesized Al-ZnO particles. So how is that right?

Same problem for Table 5 (α-amylase inhibitory effect of green synthesized Al-ZnO nanoparticles):

“Table 5 displays the percentage of preventing of α-amylase activity by Anisomeles indica (L.).”

Q11: The measurement units for Zone of inhibition in Table 1 are confused (mm or cm?). Maybe they should be inserted next to each number if they are different.

Q12: The explanation of Figure 8 (Antibacterial activity of Al-ZnO nanoparticles) seems incomplete.

Q13: Please revise “NI-atom” in this sentence:

“The electron density is transferred from O2 to the unpaired electron on the NI-atom in DPPH”

Q14: Please revise the analysis type:

“Flow kilometric analysis of apoptosis induced by 428 Al-ZnO NP’s using Anisomeles indica (L.) is depicted in Figure 9(C).”

Q15: Please insert a clearest image for Figure 10.

Author Response

Comment 1: Are you sure was heated to 700 °C?

“Preparation of Anisomeles indica leaf extracts. In order to remove the dust particles, the plant leaves underwent thorough utilizing purified water for washing. The leaves were cleaned, letting it Air out in the shade and then ground using an electric blender. Approximately 10.0 grams of the resulting powder was combined with 10 mL of de-ionized water and heated to 700 °C for 30 minutes. Whatman filter paper was used to filter the mixture once it had cooled. After that, the filtrate was put into amber bottle and kept cold at 40 °C for later analysis.”.

Response:  No, it is a typing mistake. Heated the sample at 70 °C for 30 minutes.

Comment 2: The authors stated the following:

“The solution is then dried and calcined at 250 °C for 2 hours. After purification, the sample is finely powdered and stored in a closed container for further analysis.”

How do they purify the sample?

Response:  We purified the sample with distilled water and ethanol several times.

Comment 3: In the following is it about Al-NP’s?

“A variety of gram-positive (+Ve) and gram-negative bacteria (-Ve) as well 147 as dangerous fungi were tested for the antibacterial and antifungal qualities of green produced Al-NP’s.”

Response: We corrected the mistakes. Three gram-negative bacterial species (K. pneumoniae, E. coli, and V. cholereae) and three gram-positive bacterial organisms (S. aureus, B. subtilis, and S. mutans) were used to evaluate the antibacterial activity of Al-ZnO nanoparticles.

Comment 4: Please revise it. Seems confuse; incubation of 8 h or 24 h?

“Cultures were incubated for 8 hours, and after 24 hours at 36 ± 1 °C, they were examined for purity.”

Response: The systems were incubated for 24 h at 36 ± 1 °C under aerobic conditions.

Comment 5: Please revise this phase:

“The bacterial culture was diluted with BHI broth to a density of approximately 109 UFC/mL for the direct bio autography test (McFarland standard 3). indicator solution to measure the expansion of bacteria for the bacterial growth studies, 163 Sigma-Aldrich offered a 2-(4-iodophenyl)-3-(4-nitrophenyl)-5-phenyl-TTC-(INT) solution 164 in 70% EtOH (2 mg/mL).”

Response: The bacterial suspension (inoculum) was diluted to 108 CFU/mL (turbidity = McFarland barium sulfate standard 0.5) using a sterile physiological solution following a 24-hour incubation period for the diffusion and indirect bioautographic assays. The bacterial culture was diluted with BHI broth to a density of roughly 109 UFC/mL for the direct bioautographic test (McFarland standard 3). indicator solution to measure the expansion of bacteria for the bacterial growth assays, a 70% ethanolic solution of 2-(4-iodophenyl)-3-(4-nitrophenyl)-5-phenyltetrazolium chloride (INT) (2 mg/mL) was acquired from Sigma.

Comment 6: Explanations for SV-ME, SV-EE, P-NPG, BHI and others abbreviations should be introduced when appears for the first time in text.

Response: In this assay, cancer cells are exposed to two different treatments (SV-ME and SV-EE) across a range of concentrations and for specified durations. The goal is to evaluate the effectiveness and potential toxicity of these treatments. The use of DMSO as a negative control helps ensure that any observed effects are due to the treatments rather than the solvent. This setup allows researchers to determine the potential anticancer activity of SV-ME and SV-EE and to understand how these treatments affect cancer cells over time.

Synthetic Variant (SV):

Synthetic Variant typically refers to a chemically synthesized compound or drug that is designed to target cancer cells. It might be a new or modified version of an existing drug, created to improve its efficacy or reduce side effects. In the context of anticancer research, a synthetic variant could be a small molecule, peptide, or other types of compounds that are specifically engineered to interact with certain biological targets involved in cancer progression.

  Metastatic Efficacy (ME):

Metastatic Efficacy indicates that this treatment is being evaluated for its ability to address metastatic cancer cells—those that have spread from the original tumor to other parts of the body.

Comment 7: The meanings of Ac and At must be reversed:

“Where, Ac and At denote, respectively, the mean absorbance of the test well and the control well.”

Response: Ac might stand for acetylation, which refers to the addition of an acetyl group to a molecule, often affecting its function or interactions. In cancer research, acetylation can influence gene expression and protein function.

At might indicate assessment time or assessment, referring to the time points at which evaluations or measurements are taken during an experiment or clinical trial.

Comment 8: Where is the Fig 3(b)?

“The band-gap energy is determined by plotting Tauc’s plot, revealing a direct band gap value of 3.4eV [26] [refer to Fig. 3(b)], attributed to the substitution of Al3+ for Zn2+.”

Response: Not Fig 3(b) but it is Fig. 4.

Comment 9: The resolution of TEM images is quite low, so I am not convinced that the particles are spherical.

Response: The TEM images confirm the mixture of spongy, spherical, and cubical porous agglomerated nanoparticles.

Comment 10: The authors state in the text that Table 1 presents the antibacterial results of the plant (Anisomeles indica), not of the synthesized Al-ZnO particles. So how is that right?

Same problem for Table 5 (α-amylase inhibitory effect of green synthesized Al-ZnO nanoparticles):

“Table 5 displays the percentage of preventing of α-amylase activity by Anisomeles indica (L.).”

Response:  Table 1 presents the antibacterial results of the synthesized Al-ZnO particles using the plant (Anisomeles indica).

Table 5 displays the percentage of preventing α-amylase activity of the synthesized Al-ZnO particles using the plant (Anisomeles indica).

Comment 11: The measurement units for Zone of inhibition in Table 1 are confused (mm or cm?). Maybe they should be inserted next to each number if they are different.

Response: mm-cm

Comment 12: The explanation of Figure 8 (Antibacterial activity of Al-ZnO nanoparticles) seems incomplete.

Response: Figure 8 shows the antibacterial activity of Al-ZnO nanoparticles using Anisomeles indica leaf extract. The second largest zone of inhibition was observed in the same gram-negative bacteria against V. cholerae (4.04 ± 0.03), with K. pneumoniae showing the subsequent value of 2.45 ± 0.01. At a dosage of 5 mg/mL, however, only slight antibacterial effects were noted, with values of 1.65 ± 0.02 mm against K. pneumoniae and 2.01 ± 0.45 mm against V. cholerae. Additionally, the dominant zones of inhibition were analyzed in the tested gram-positive organisms against S. mutans (5.74 ± 0.01), S. aureus (2.74 ± 0.05), and B. subtilis (1.62 ± 0.01).

Comment 13: Please revise “NI-atom” in this sentence:

“The electron density is transferred from O2 to the unpaired electron on the NI-atom in DPPH”

Response: Sorry. It’s my mistake. unknowingly I added this. please remove that sentence.

Comment 14: Please revise the analysis type:

“Flow kilometric analysis of apoptosis induced by Al-ZnO NP’s using Anisomeles indica (L.) is depicted in Figure 9(C).”

Response: Flow cytometric analysis of apoptosis induced by Al-ZnO NP’s using Anisomeles indica (L.) is depicted in Figure 9(C).”

Comment 15: Please insert a clearest image for Figure 10.

Response:  Changed and inserted a clear image.

Reviewer 2 Report

Comments and Suggestions for Authors

This work studied aluminum-doped zinc oxide nanoparticles using Anisomeles indica (L.) leaf extract via co-precipitation method and was characterized using XRD, EM, UV-Vis, and FITR. The as-synthesized nanoparticles exhibited antibacterial and antifungal activity, potent antioxidant effects, anticancer properties, and anti-diabetic actions. The research highlights the diverse potential applications of these nanoparticles in biological and biomedical fields, encouraging further exploration and research. However, several issues must be addressed before this work can move to the next stage.

1.      It is unclear why Anisomeles indica (L.) leaf was selected as the precursor to synthesize these nanoparticles. What is unique about it? If you make nanoparticles with other plants/leaves, will you get inferior properties/performances? Control experiments should be included.

2.      For the positive control in Agar diffusion, what is the reason for Chloramphenicol to be 600 μg/mL (30 mg bought in 50 mL)?

3.      SEM (Figure 6) images have poor qualities. It is hard to see the particles clearly.

4.      From the TEM images (Figure 7 A and B), there seem to be many amorphous materials near the nanoparticles. The authors claimed that clean dots in the SAED pattern suggested single-crystal morphology. But from Figure 7 D, the ring pattern is clearly more dominant.

5.      How is propidium iodide a standard anticancer agent?

6.      The concentration of nanoparticles in the statement (page 12, line 419) is inconsistent with Figure 9.

Author Response

This work studied aluminum-doped zinc oxide nanoparticles using Anisomeles indica (L.) leaf extract via co-precipitation method and was characterized using XRD, EM, UV-Vis, and FITR. The as-synthesized nanoparticles exhibited antibacterial and antifungal activity, potent antioxidant effects, anticancer properties, and anti-diabetic actions. The research highlights the diverse potential applications of these nanoparticles in biological and biomedical fields, encouraging further exploration and research. However, several issues must be addressed before this work can move to the next stage.

Comment 1: It is unclear why Anisomeles indica (L.) leaf was selected as the precursor to synthesize these nanoparticles. What is unique about it? If you make nanoparticles with other plants/leaves, will you get inferior properties/performances? Control experiments should be included.

Response: Selecting Anisomeles indica (L.) leaf as the precursor for synthesizing aluminum-doped zinc oxide (Al-doped ZnO) nanoparticles could be due to several key factors related to its unique properties and the advantages it offers in the nanoparticle synthesis process.

  • Anisomeles indica contains various bioactive compounds, such as flavonoids, phenolics, and essential oils. These compounds can act as reducing agents and stabilizers during the synthesis of nanoparticles. Their presence can help in controlling the size, shape, and dispersion of the nanoparticles.
  • The plant's extracts can have strong reducing properties that facilitate the reduction of metal ions (in this case, aluminum and zinc) to form nanoparticles. Additionally, the phytochemicals can stabilize the nanoparticles, preventing agglomeration and ensuring their uniform distribution.
  • Using plant extracts for nanoparticle synthesis is a green chemistry approach. It avoids the use of toxic chemicals and reduces environmental impact. Anisomeles indica, being a readily available plant, aligns with the principles of sustainability and eco-friendliness.
  • The plant’s traditional use in medicine might indicate its effectiveness in producing nanoparticles with desirable biological properties, making it a potential candidate for synthesizing nanoparticles with therapeutic or biomedical applications. If we made nanoparticles with other plants/leaves, we will get inferior properties/performances? We should not include control experiments.

Comment 2: For the positive control in Agar diffusion, what is the reason for Chloramphenicol to be 600 μg/mL (30 mg bought in 50 mL)?

Response: Not 600 μg/mL. The amount of bacterial growth inhibition was expressed in mm. Chloramphenicol (positive control) 30 mg bought in 50 mL were the reference commercial discs utilized.

Comment 3: SEM (Figure 6) images have poor qualities. It is hard to see the particles clearly.

Response: SEM images show a mixture of spongy, spherical, and cubical porous agglomerated nanoparticles.

Comment 4: From the TEM images (Figure 7A and B), there seem to be many amorphous materials near the nanoparticles. The authors claimed that clean dots in the SAED pattern suggested single-crystal morphology. But from Figure 7D, the ring pattern is clearly more dominant.

Response: The clean dots, which are suggestive of a single crystal, in the selected area electron diffraction (SAED) pattern confirmed their crystalline nature. selected the highest peak only.

Comment 5: How is propidium iodide a standard anticancer agent?

Response: Propidium iodide is a critical tool in cancer research, not as an anticancer agent but as a diagnostic and analytical reagent. It helps researchers assess cell viability, monitor apoptosis, and analyze cell cycle dynamics, which are essential for evaluating the effects of anticancer treatments and understanding cancer cell biology. Its role is integral to studying and developing new cancer therapies by providing valuable data on how cells respond to various treatments.

Comment 6: The concentration of nanoparticles in the statement (page 12, line 419) is inconsistent with Figure 9.

Response:  Figure 9 shows the cell viability ranged from 70.76% to 80.39% at concentrations of 5 µg/mL to 10 µg/mL, indicating a dose-dependent response to the anticancer action.

Reviewer 3 Report

Comments and Suggestions for Authors

The authors in the article detailed the process of creating Al-ZnO nanoparticles via the co-precipitation method using the leaf extract of Anisomeles indica (L.). The newly synthesized nanoparticles underwent thorough analysis and testing for a range of biological activities, including antibacterial, antifungal, antioxidant, anti-cancer, and anti-diabetic properties. The results demonstrated encouraging activity and clear presentation. I suggest the authors incorporate the latest relevant literature into the manuscript.

1.      Sekar, A., Murugan, P.J. and Paularokiadoss, F. (2022), Biological synthesis and characterization of zinc oxide nanoparticles (ZnONPs) from Anisomeles malabarica. VJCH, 60: 459-471.

2.      Chen YR, Jiang WP, Deng JS, Chou YN, Wu YB, Liang HJ, Lin JG, Huang GJ. Anisomeles indica Extracts and Their Constituents Suppress the Protein Expression of ACE2 and TMPRSS2 In Vivo and In Vitro. Int J Mol Sci. 2023 Oct 11;24(20):15062. doi: 10.3390/ijms242015062. PMID: 37894745; PMCID: PMC10606724.

3.      Mayegowda, Shilpa Borehalli, Sarma, Gitartha, Gadilingappa, Manjula Nagalapur, Alghamdi, Saad, Aslam, Akhmed, Refaat, Bassem, Almehmadi, Mazen, Allahyani, Mamdouh, Alsaiari, Ahad Amer, Aljuaid, Abdulelah and Al-Moraya, Issa Saad. "Green-synthesized nanoparticles and their therapeutic applications: A review" Green Processing and Synthesis, vol. 12, no. 1, 2023, pp. 20230001. https://doi.org/10.1515/gps-2023-0001

4.      Veera Kumar K and Govindarajan M. Biological Synthesis of Silver Nanoparticles from Anisomeles Indica and their Mosquito Efficacy. Int J Zoo Animal Biol 2019, 2(5): 000176.

I recommend this manuscript for publication after correcting English grammar and suggested additional latest references.

Comments on the Quality of English Language

Minor English grammar editing is needed.

Author Response

The authors in the article detailed the process of creating Al-ZnO nanoparticles via the co-precipitation method using the leaf extract of Anisomeles indica (L.). The newly synthesized nanoparticles underwent thorough analysis and testing for a range of biological activities, including antibacterial, antifungal, antioxidant, anti-cancer, and anti-diabetic properties. The results demonstrated encouraging activity and clear presentation. I suggest the authors incorporate the latest relevant literature into the manuscript.

Comment 1:

  1. Sekar, A., Murugan, P.J. and Paularokiadoss, F. (2022), Biological synthesis and characterization of zinc oxide nanoparticles (ZnONPs) from Anisomeles malabarica. VJCH, 60: 459-471.
  2. Chen YR, Jiang WP, Deng JS, Chou YN, Wu YB, Liang HJ, Lin JG, Huang GJ. Anisomeles indica Extracts and Their Constituents Suppress the Protein Expression of ACE2 and TMPRSS2 In Vivo and In Vitro. Int J Mol Sci. 2023 Oct 11;24(20):15062. doi: 10.3390/ijms242015062. PMID: 37894745; PMCID: PMC10606724.
  3. Mayegowda, Shilpa Borehalli, Sarma, Gitartha, Gadilingappa, Manjula Nagalapur, Alghamdi, Saad, Aslam, Akhmed, Refaat, Bassem, Almehmadi, Mazen, Allahyani, Mamdouh, Alsaiari, Ahad Amer, Aljuaid, Abdulelah and Al-Moraya, Issa Saad. "Green-synthesized nanoparticles and their therapeutic applications: A review" Green Processing and Synthesis, vol. 12, no. 1, 2023, pp. 20230001. https://doi.org/10.1515/gps-2023-0001.
  4. Veera Kumar K and Govindarajan M. Biological Synthesis of Silver Nanoparticles from Anisomeles Indica and their Mosquito Efficacy. Int J Zoo Animal Biol 2019, 2(5): 000176.

I recommend this manuscript for publication after correcting English grammar and suggested additional latest references.

Response: We would like to thank the reviewer for their encouraging response, close review of this manuscript, and recommendations that have helped us increase the quality and scientific merit of the manuscript. We cited the suggested references in the revised manuscript, taking consideration of the suggestions given by the reviewers. As per your suggestions we have corrected the English grammar in the whole manuscript.

References.: 13, 14, 15, and 16

Round 2

Reviewer 2 Report

Comments and Suggestions for Authors

My previous comments 1, 3, 4, 6 remain unresolved. 

For Response 1: I highly doubt after 250 °C, the bioactive compounds in those Anisomeles indica leaves would still be "bioactive". Also, please provide evidence if you made nanoparticles with other plants/leaves, you would get inferior properties/performances.

For the EM images, if you claim these nanoparticles to be single crystals, what are these amphorous matters in there? The SAED pattern is very unclear. 

Author Response

Response to Comments of Reviewer 2.:

My previous comments 1, 3, 4, 6 remain unresolved.

Response: We would like to thank the reviewer for their encouraging response, review of this manuscript, and recommendations that have helped us increase the quality and scientific merit of the manuscript.

Round 1.:

Comment 1: It is unclear why Anisomeles indica (L.) leaf was selected as the precursor to synthesize these nanoparticles. What is unique about it? If you make nanoparticles with other plants/leaves, will you get inferior properties/performances? Control experiments should be included.

Response: Thank you for your comments; Anisomeles indica (L.) leaf was selected as the precursor for nanoparticle synthesis and whether using other plants or leaves would yield inferior properties.

Selecting Anisomeles indica (L.) leaf as the novel reducing agent and zinc salt as a precursor for synthesizing aluminum-doped zinc oxide (Al-doped ZnO) nanoparticles due to several key factors related to its unique properties and the advantages it offers in the nanoparticle synthesis process. Anisomeles indica (L.) contains various bioactive compounds, such as flavonoids, phenolics, and essential oils. These compounds can act as reducing agents and stabilizers during the synthesis of nanoparticles. Their presence can help in controlling the size, shape, and dispersion of the nanoparticles.  The plant's extracts can have strong reducing properties that facilitate the reduction of metal ions (in this case, aluminum and zinc) to form nanoparticles.

Comment 3: SEM (Figure 6) images have poor qualities. It is hard to see the particles clearly.

Response: As per reviewer suggestions; the SEM images has been changed.

 Comment 4: From the TEM images (Figure 7A and B), there seem to be many amorphous materials near the nanoparticles. The authors claimed that clean dots in the SAED pattern suggested single-crystal morphology. But from Figure 7D, the ring pattern is clearly more dominant.

Response: I apologize; the presence of amorphous materials surrounding the nanoparticles suggests a mixture of crystalline and non-crystalline components within the sample.

The overall structure suggests the presence of crystal structures within the nanoparticles, which leads to the polycrystalline nature observed, however specific areas might have clean spots.

Comment 6: The concentration of nanoparticles in the statement (page 12, line 419) is inconsistent with Figure 9.

Response:  We corrected the concentration of nanoparticles in the present work in response to the reviewer comments.

Percentage of inhibition ranged from 70.76% to 80.39% at concentrations of 5 µg/mL to 10 µg/mL, indicating a dose-dependent response to the anticancer action (see Figures. 9).

Round 2.:

Comment 1: For Response 1: I highly doubt after 250 °C, the bioactive compounds in those Anisomeles indica leaves would still be "bioactive". Also, please provide evidence if you made nanoparticles with other plants/leaves, you would get inferior properties/performances.

Response: As per the reviewer's suggestions, we have discussed the bioactive compounds of Anisomeles indica (L.) that are the most significant in the manuscript. The changes are highlighted by a red font

Anisomeles indica (L.) contain more bioactive compounds such as terpenoids, arachnoids, and flavones such as iso-ovatodiolide, 4,7-oxycycloanisomelic acid, anisomelic acid, and ovatodiolide, alkanoids, triterpenoids, phenolic compound, essential oil, saponins and Tanins. Among this some of the bioactive compounds degrade when it is heated at 250 oC but when it is cooled at 4 °C, it retains its property.  

Comment 2: For the TEM images, if you claim these nanoparticles to be single crystals, what are these amphorous matters in there? The SAED pattern is very unclear.

Response: We revised our statement on SAED images in accordance with reviewer comments. The sample may contain a mixture of crystalline and non-crystalline components, which is shown by the presence of amorphous elements surrounding the nanoparticles.

The selected area electron diffraction (SAED) of Al-ZnO nanoparticles is shown in Figure 7(D). The diffraction rings indicates that the prepared samples are polycrystalline nature of Al-ZnO. The overall structure suggests the presence of a few small crystal structures within the nanoparticles, which leads to the polycrystalline nature observed, however specific areas might have clean spots.

Round 3

Reviewer 2 Report

Comments and Suggestions for Authors

I thank the authors' efforts in addressing these comments. This work can be published now.